# Association between the Genetic Variants of Glutathione Peroxidase 4 and Severity of Endometriosis

**DOI:** 10.3390/ijerph17145089

**Published:** 2020-07-15

**Authors:** Yun-Yao Huang, Cheng-Hsuan Wu, Chung-Hsien Liu, Shun-Fa Yang, Po-Hui Wang, Long-Yao Lin, Tsung-Hsien Lee, Maw-Sheng Lee

**Affiliations:** 1Department of Obstetrics and Gynecology, Chung Shan Medical University Hospital, Taichung 40203, Taiwan; xerospin@gmail.com (Y.-Y.H.); chliu@csmu.edu.tw (C.-H.L.); wang082160@gmail.com (P.-H.W.); xillin681113@gmail.com (L.-Y.L.); msleephd@gmail.com (M.-S.L.); 2Women’s Health Research Laboratory, Changhua Christian Hospital, Changhua 50006, Taiwan; 97528@cch.org.tw; 3School of Medicine, Kaohsiung Medical University, Kaohsiung 80708, Taiwan; 4School of Medicine, Chung Shan Medical University, Taichung 40203, Taiwan; 5Institute of Medicine, Chung Shan Medical University, Taichung 40203, Taiwan; ysf@csmu.edu.tw; 6Department of Medical Research, Chung Shan Medical University Hospital, Taichung 40203, Taiwan; 7Division of Infertility, Lee Women’s Hospital, Taichung 40602, Taiwan

**Keywords:** single nucleotide polymorphism, oxidative stress, endometriosis, glutathione peroxidase, thioredoxin, thioredoxin reductase

## Abstract

It has been reported that oxidative and nitrative stress might be the pathogenesis of endometriosis. This prospective case-control study attempted to check the connection between single nucleotide polymorphism (SNP) of three antioxidant enzymes (glutathione peroxidase 4 (GPX4), thioredoxin 2 (TXN2), thioredoxin reductase 1 (TXNRD1)) and endometriosis. We recruited 90 patients with histology-approved endometriosis as the case group and 130 age-matched women for an annual pap smear examination as the control group. The stage of endometriosis was evaluated with revised ASRM score. Both groups were genotyped in the peripheral leukocytes for the SNP of GPX4 (rs713041), TXN2 (rs4821494) and TXNRD1 (rs1128446) by PCR-based methods. An X^2^ test was used to analysis of the difference of allele frequency and SNP distribution between two groups. The results revealed GPX4 (rs713041) has a significantly different distribution between two groups (C:T = 116 (44.6%):144 (55.4%) in control and C:T = 104 (57.8%): 76 (42.2%) in endometriosis groups, *p* = 0.007). The SNP in TXN2 (rs4821494) also showed a difference in allele frequency (G:T = 180 (69.2%):80 (30.8%) in control and G:T = 141 (78.3%):39 (21.6%) in endometriosis group, *p* = 0.030). In addition, the SNP GPX4 (rs713041) was associated with the severity of the endometriosis. Women who have advanced stage endometriosis were different from mild endometriosis in genetic variants of GPX4 gene (*p* = 0.001). In conclusion, the relationship between endometriosis and SNP of antioxidant enzymes, GPX4 and TXN2, was confirmed by the present study. According to the result, we suggested that the GPX4 might contribute to the pathogenesis of endometriosis.

## 1. Introduction

Endometriosis is a debilitating disease characterized by the implantation of endometrial tissue outside the uterine cavity. As many as 10–15% of women of reproductive age are affected by this disease [1]. Ballweg et al. in 2011 reported the average waiting period for a definitive diagnosis of endometriosis required as long as nine years [2]. While a definitive diagnosis can be confirmed through laparoscopy in more severe cases, for milder cases, especially in teenagers and younger women, it remains difficult, as there are currently no reliable biomarkers available. It is generally accepted that the disease began by the implantation of endometrial cells through retrograde menstruation [3], but this theory cannot fully explain why only 10% of women with retrograde menstruation develop the disease.

Recent attention has been focused on the effect of oxidative stress in the development of endometriosis. The glutathione (GSH) system is one of the two major thiol-dependent antioxidant systems in mammalian cells, that participates in the defense against oxidative stress via the removal of various ROS by glutathione peroxidase (GPX). Glutathione peroxidase exists in at least five isoforms in the human body, GPX1-5 [4]. Of these five types, GPX4 protects cells against membrane peroxidation and cell death by catalyzing the reduction of hydrogen peroxide and lipid peroxides at the expense of reduced glutathione, thus protecting cells from oxidative stress [5,6]. The gene for GPX4 is located on chromosome 19, it is a selenium-containing enzyme, where selenium is incorporated into the primary structure as the 21st amino acid selenocysteine (Secys). Gene variation in the selenoprotein genes may influence the efficiency of this enzyme [7,8]. The rs713041 SNP causes a C-T substitution in the GPX4 gene in a region corresponding to the 3′untranslated region (3′UTR) of the mRNA, in turn altering the protein binding to the 3′UTR. This region is crucial for selenoproteins, as it is required for incorporation of Secys; without it, the translation would stop at the UGA codon, resulting in a non-functional protein [9]. Any SNPs at this region could potentially effect gene activity and enzyme function [9,10]. At a low selenium intake, SNPs at this region may influence the susceptibility to disease [8] GPX4 SNP (rs713041) had been demonstrated in previous studies to be associated with an increased risk of death in breast cancer patients. There is also evidence that a higher risk of colorectal cancers is observed in altered SNP (C-T) of the GPX4 gene [11]. In the field of infertility, Khadzhieva et al. investigated the role of oxidative stress in recurrent miscarriages with unknown etiology [12]. GPX4 (rs713041) along with several other alleles were shown to be risk alleles for idiopathic recurrent miscarriage. Furthermore, Ota et al. [13] reported a different expression of glutathione peroxidase between eutopic and ectopic endometrial tissues. There was a phase-dependent change of GPX expression in the controls, but a persistent expression in the endometriosis group. This suggests a difference in genotypes of GPX4 (rs713041) may play a role in pathogenesis of endometriosis.

The thioredoxin system is the other major thiol-dependent antioxidant system in mammalian cells, this disulfide reductase system includes NADPH, thioredoxin reductase (TXNRD) and thioredoxin (TXN). The main function of these enzymes is to transfer electrons through the dithiol-difulfide exchange reaction, in order to defend against oxidative stress and is also a crucial step in DNA synthesis [14]. The mammalian cells possess 2 TXN systems, TXN1 which is found in cytosol, and TXN2 in mitochondria [15]. The overexpression of TXN and [16,17] TXNRD1 has been reported in various types of organ cancers including gastric, lung, cervical and breast cancer [18]. The overexpression of TXN and TXNRD improved the survival of cancer cells as it decreases apoptosis and enhances its tolerance to oxidative stress. Extending previous research, we hypothesize that endometriosis patients may also have an overexpression of TXN genes, allowing ectopic endometrial tissue to proliferate. However, limited research is found between the TXN gene and endometriosis. Choi et al. [19] compared follicular fluid levels of thioredoxin in endometriosis and control patients receiving IVF. Levels of all the inflammatory cytokines positively correlated with the levels of TXN but did not reach statistical significance. Seo et al. [20] used rt-PCR to compare TXN and TXN binding protein 2 mRNA levels in endometrium of patients with endometriosis. No significant difference was found between the groups. This study focused on a SNP of the TXN2 gene (rs4821494) and a SNP of the TXNRD1 (rs1128446), both have been reported to be overexpressed by cancer cells, but currently no strong evidence in relation to endometriosis.

It is currently unknown whether the genetic polymorphisms of these oxidative stress enzymes play a role in development of endometriosis. In an attempt to answer this question, this study compared the genetic variants of three different oxidative stress enzymes (GPX4, TXN2 and TXNRD1) and evaluated if there is a difference in genotype and allele frequency between women with endometriosis and healthy controls.

## 2. Materials and Methods

### 2.1. Study Design and Subsjects

This study is a prospective case-control study where we compare distribution of selected SNPs in oxidative enzymes GPX4, TXN2 and TXNRD1 from endometriosis patients and healthy controls. We recruited 90 patients with endometriosis, who underwent diagnostic or surgical laparoscopy to examine the pelvic organs and/or remove endometriosis lesions. A venous blood sample was collected for DNA extraction and genotyping from cases undergoing laparoscopy for endometriosis, in the morning after surgery, at Chung Shan medical University hospital from April 2011 to April 2015. In addition, a total of 130 age-matched control cases had a blood sample taken during their health check for annual pap smear at the same hospital. Participants of the control group had no previous record of chronic pelvic pain, dysmenorrhea, or dyspareunia. Control cases had blood sample taken during their health check for annual pap smear at the hospital. The institutional review board of Chung Shan Medical University Hospital approved this study, and each patient signed informed consent before participating in the study. Basic information such as clinical diagnosis, age, obstetric history, height, weight, dysmenorrhea severity (from 0 = none for pain score 0, 1 = mild for pain score 1–4, 2 = severe for pain score 5–9), sonography reports, medications used before and after surgery and surgical staging (rASRM score) were collected for all patients about to receive laparoscopy to investigate for causes of infertility, chronic pelvic pain, adnexal mass or suspicious pelvic endometriosis. The ages of the control group were obtained from data collected. To keep congruent data only patients with Han Chinese descents were recruited for this study.

### 2.2. DNA Extraction and Determination of Genotypes

The following single nucleotide polymorphism (SNPs) GPX4 (rs713041), TXN2 (rs4821494), TXNRD1 (rs1128446) were evaluated by using the Taqman allelic discrimination assay (TaqMAn SNP). The SNPs were chosen based on the international HapMap project (http://hapmap.ncbi.nlm.nih.gov), and searches in the dbSNP (http://www.ncbi.nlm.nih.gov/snp) as the work by Kevenaar et al. [21]. Genomic DNA was extracted from EDTA anti-coagulated venous blood using a QIAamp DNA blood mini kit (Qiagen, Valencia, CA, USA), according to the manufacturer’s instructions described in detail previously [22]. We dissolved the DNA ins TE buffer (10 mM Tris and 1 mM EDTA acid; pH 7.8) and then measured the optical density at 260 nm to determine the DNA quantity. The final preparation was stored at −20 °C and used as templates for polymerase chain reaction (PCR). Allele discrimination of the three studied SNPs was assessed with the ABI StepOne™ Real-Time PCR System (Applied Biosystems, Foster City, CA, USA) and analyzed using SDS version 3.0 software (Applied Biosystems), with the TaqMan assay [23].

### 2.3. Statistical Analysis

The Hardy-Weinberg equation was used to calculate the expected numbers and then compared with the actual numbers of each genotype, including GPX4 (rs713041), TXN2 (rs4821494), and TXNRD1 (rs1128446). A chi-square test was performed to determine the Hardy–Weinberg equilibrium. The associations were examined between tested SNPs and endometriosis under the different genetic models: genotypic model (AA versus Aa versus aa), recessive model (AA versus Aa + aa), and allele frequency.

The demographic data and other clinically relevant data of continuous variables are presented as means (standard deviation, SD) after the Kolmogorov–Smirnov test for normal distribution, whereas categorical variables are presented as numbers and percentages. Differences were compared between groups using the Mann–Whitney U test (for continuous variables) or chi-square test (for categorical variables) when appropriate. Then qualitative variables across groups were compared by the Chi-square square test to see if there is a high-risk genotype for endometriosis. Finally, endometriosis cases were separated into two groups mild (stage I, II) and advance (stage III, IV), according to the rASRM score for comparison between the selected SNPs of oxidative enzymes and whether they are related to disease severity. All data were analyzed using the IBM SPSS Statistics for Windows, Version 22.0 (IBM Corp., Armonk, NY, USA). *p*-values < 0.05 were considered statistically significant.

## 3. Results

A total of 220 women were included in this prospective case-control study. Ninety women with endometriosis, confirmed by laparoscopy, and 130 age-matched women in the health examination program were recruited as the control group. All the participants in the control group had no previous medical record of chronic pelvic pain, dysmenorrhea, or dyspareunia. The average age of the endometriosis group is 35.09 ± 7.48 years old, and the average age of control group is 36.76 ± 6.84 years old (*p* = 0.115, Student’s *t*-test). The demographic characteristics of the patients with endometriosis are listed in Table 1. Using the revised American Society for reproductive Medicine classification of endometriosis, rASRM score), 13 (14.4%) had stage I endometriosis, 27 (30%) had stage II endometriosis, 38 (42.2%) had stage III endometriosis and 12 (13.3%) had stage IV endometriosis. In a self-reported pain score of dysmenorrhea, ranging from 0 to 9, 25 (27.8%) reported 0, 18 (20%) reported 1–4 and 47 (52.2%) reported 5–9 of pain scores.

The Hardy–Weinberg equilibrium (HWE) test was utilized to detect the distribution of the SNP alleles GPX4 (rs713041), TXN2 (rs4821494), and TXNRD1 (rs1128446) in the studied patients. Both GPX4 (rs713041) and TXN2 (rs4821494) genes were confirmed by HWE test in the case and control groups, demonstrating the hereditary distribution of both SNP alleles is compatible with Mendelian genetic law. However, the TXNRD1 (rs1128446) gene was not tested here due to genotype G/G being absent in both case and control groups.

Table 2 summarizes the frequencies of GPX4 (rs713041) polymorphism among women with endometriosis or normal control group. In the codominant genotypic test model, the CC, CT and TT genotype frequencies of GPX4 (rs713041) were 33 (36.7%), 38 (42.2%) and 19 (21.2%) in endometriosis cases, and 22 (16.9%), 72 (55.4%), 36 (27.7%) in the control group, respectively (*p* = 0.004, *X*^2^ test). We observed a significant difference in CC genotype frequencies between endometriosis patients and control cases in this codominant genotypic test model 36.7% and 16.9%, respectively (*p* = 0.004). In the recessive test model, the CC and CT/TT genotype frequencies of GPX4 (rs713041) were 33 (36.7%), and 57 (63.3%) in endometriosis cases, and 22 (16.9%), and 108 (83.1%) in the control group, respectively (*p* = 0.001, *X*^2^ test). The C allele frequency is 104 (57.8%) in endometriosis cases, compared to 116 (44.6%) in control cases. The T allele frequency was 76 (42.2%) in endometriosis cases compared to 144 (55.4%) in control cases (*p* = 0.007).

In Table 3 we compare frequencies of TXN2 (rs4821494) polymorphism among women with or without endometriosis. In the codominant model, the GG, GT and TT genotype frequencies of TXN2 (rs4821494) were 54 (60%), 33 (36.7%) and 3 (3.3%) in endometriosis group and 60 (46.2%), 60 (42.2%) and 10 (7.7%) in the control group. Genotype GG in the codominant model also showed a higher frequency in endometriosis cases 54 (60%) compared to controls 60 (46.2%, *p* = 0.09). In the recessive model, the GG genotype frequency was 54 (60%) in endometriosis cases, and 60 (46.2%) in the control group, *p* = 0.043. The GT/TT genotype frequency was 36 (40%) in endometriosis cases and 70 (53.8%) of control cases. The Allele G frequency was found in 141 (78.3%) of endometriosis cases, and 180 (69.2% of control cases, *p* = 0.035. The T allele frequency was 39 (21.6%) of endometriosis cases and 80 (30.8%) of control cases.

Table 4 compares the frequencies of TXNRD1 (rs1128446) polymorphism among women with or without endometriosis. In the codominant genotypic test model, the CC, CG and GG genotype frequencies of TXNRD1 (rs1128446) were 89 (98.9%), 1 (1.1%) and 0 in endometriosis cases, and 125 (96.2%), 5 (3.8%), 0 (2.6%) in the control group, respectively, but did not reach a significant statistical difference (*p* = 0.221). In the recessive test model, the CC and CG genotype frequencies were 89 (98.9%) and 1 (1.1%) in the endometriosis cases and 125 (96.2%) and 5 (3.8%) in the control cases, which also did not reach statistical significance. The Allele C frequency was 179 (99.4%) and 235 (98.1%), respectively, in endometriosis and control cases. The Allele G is almost undetectable in this population, found only in 1 (0.6%) and 5 (1.9%) of the control cases.

Table 5 demonstrated the comparison of genotype distribution of GPX4 (rs713041) and TXN2 (rs4821494) between the patients of rASRM stage 1–2 (minimal–mild) endometriosis with the patients of advanced stage 3–4 endometriosis. In the CC group of the GPX4 (rs713041), we observed a higher frequency in the advanced stage 3–4 patients (26 (56%)) than that in stage 1–2 patients (7 (17.5%), *p* = 0.001). No significant difference was found between stage 1–2 versus 3–4 endometriosis patients in the TXN2 (rs4821494) SNP. Because the C:G allele of TXNRD1 (rs1128446) is 179:1 in patients with endometriosis, the comparison between stage of the disease is not practical for biostatistics. Consequently, frequencies of TXNRD1 (rs1128446) polymorphism was not shown in Table 5.

## 4. Discussion

In this study, we compared the genetic variation of three antioxidant enzymes, GPX4, TXN2 and TXNRD1 and attempted to identify whether there is any difference in allele frequency between endometriosis and control patients. Regarding the GPX4 (rs713041) SNP, there was a significant difference in allele frequency between endometriosis and control groups and between the early and advanced stages of endometriosis groups. TXN2 (rs4821494) SNP showed a difference in allele frequency, but no difference in the severity of endometriosis genotype polymorphism. There was no difference of allele frequency for TXNRD1 (rs1128446) SNP in both groups.

It has been reported that GPX is expressed differently in eutopic and ectopic endometrial tissue [13]. Ota et al. found there was phase-dependent changes of GPX expression in the surface and glandular epithelia in the eutopic endometrium of the fertile controls. However, in endometriosis cases, the expression of GPX lost this variation during the menstrual cycle. The expression of GPX in adenomyosis was persistently elevated over the control levels throughout the menstrual cycle, suggesting a pathological role in endometriosis [13]. Our results found a difference in allele frequencies between endometriosis and healthy controls. Our results suggest that a difference in rs713041 (CC vs. CT + TT) of GPX4 may be a reason for the occurrence and severity of endometriosis. Polak et al. [24] assessed the concentration of plasma GPX in the peritoneal fluid of patients with unexplained infertility and minimal–mild endometriosis. They found the unexplained infertility group had lower peritoneal GPX concentration than control group and no difference between mild endometriosis to control. While a comparison of GPX concentration between mild endometriosis and controls is not available from our study, we found that there is a significant difference of allele frequency in GPX4 rs713041 SNP between early (stage I–II) and advanced (stage III–IV) stages of endometriosis. Patients with advanced stage endometriosis featured a higher frequency in the CC genotype of GPX4 rs713041 than that of stage I-II patients. The findings of our study indicated that genotype GPX4 rs713041 may be associated with not only the occurrence but also the severity of endometriosis. Since the GPX4 rs713041 SNP is associated with mortality of breast cancer patients and the risk of colorectal cancer [11], the CC allele of GPX4 rs713041 SNP may increase the survival of cancer cells or ectopic endometrium by against the oxidative stress. Consequently, the CC allele is related to the occurrence and severity of endometriosis.

The other enzyme SNP this study focused on is the SNP rs4821494 of TXN2 gene, which was located on chromosome 22. The thioredoxin and thioredoxin receptor has been reported to be overexpressed in many cancer cells, as it allows for cancer cells to defend themselves against oxidative stress, thus decreasing in apoptosis [25]. The overexpression had been reported in gastric cancer, lung cancer, renal cell carcinoma, cervical cancer and breast cancer [25,26,27]. It is probable to hypothesize endometriosis patients may also have overexpression of TXN genes, to allow for ectopic endometrial tissue to survive against oxidative stress. However, only a few studies investigate the association between TXN and endometriosis. Seo et al. [20] found no significant differences in thioredoxin mRNA levels in the endometrium of patients with endometriosis and the control groups. However, Thioredoxin-binding protein-2 (TBP-2) mRNA levels in the endometrium were lower, and the TRX to TBP-2 ratio was higher in patients with endometriosis than in the control group. They then deduced the possibility of a mutual disproportion of TRX and TBP-2 in patients with endometriosis, leading to TBP-2 downregulation, which in turn negatively regulates TXN, decreasing its expression. No significant risk of endometriosis by genetic variants was found in the TXN (rs4821494) in a codominant model. However, we observe that the GG genotype is 1.75 times more likely to appear in endometriosis patients in recessive model of genetic variant analysis.

In order to further explore the role of the thioredoxin system and its impact on endometriosis, we chose to analyze SNP (rs1128446) of TXNRD1 gene, located on chromosome 12. No significant risk of endometriosis by genotype was found in the TXNRD1 (rs1128446) through this study. We found no difference in SNP rs1128446 of TXNRD1 gene, both in genotypes and allele frequencies between endometriosis and controls. However, an interesting observation was made, as only two genotypes (CC, CG) exist in the TXNRD1 gene (rs1128446), and no GG genotype was identified in the present study. To explain this phenomenon, we searched the national center for biotechnology information (NCBI) database SNP, and genotype G was found in only 18.81% of the population. This is even lower in the Asian population, with Han Chinese at 3.4%, and Japanese 0%. Therefore, it is reasonable to find the absence of the GG genotype in our study (www.ncbi.nlm.nih.gov/snp/?term=rs1128446).

Our study is, to the best of our knowledge, the first study to use SNP genotypic assay to compare the distribution of the SNP rs713041 alleles on GPX4 gene between endometriosis patients and healthy controls in a Han population. Previous studies regarding endometriosis and GPX4 relations composed mostly quantitative protein concentration measurement found in serum, peritoneal fluids or ectopic endometrial tissues [28]. Our study was able to provide a reason as to why there were different levels of concentrations of GPX4 enzyme between healthy and infertile women with endometriosis, as the genotype polymorphism rs713041 may have led to altered structure or function of GPX4 protein.

There are several limitations to this study. One of the limitations is the uncertainty that members of the control group were totally disease free, as they did not go through laparoscopic surgeries to confirm for presence or absence of endometriosis, but rather are self-reported to have no symptoms of the disease. Nonetheless, even if the control group has some women with occult endometriosis, the difference between endometriosis and endometriosis-free patients in the allele frequency for SNP rs713041 of GPX4 is still significant.

In conclusion, our findings imply that the genetic polymorphism of the GPX4 gene likely contributes to the pathogenesis of endometriosis. Further research with a larger sample size and preferably immunnohistological stains for GPX4 and TXN2 expression is warranted to understand their roles in development of endometriosis.

## Figures and Tables

**Table 1 ijerph-17-05089-t001:** Demographic data of endometriosis patients.

Variables	Case	Mean ± S.D. or Percentage
Age (years)	90	35.09 ± 7.48
BMI	90	22.75 ± 4.1
rASRM stage		
Minimal	13	14.4%
Mild	27	30.0%
Moderate	38	42.2%
Severe	12	13.3%
Pain score		
0	25	27.8%
1–4	18	20.0%
5–9	47	52.2%

**Table 2 ijerph-17-05089-t002:** Frequencies of GPX4 (rs713041) polymorphism among women with endometriosis or normal control group.

GPX4 (rs713041) Genotype		Endometriosis				
		Negative		Positive		
		N	%	N	%	*p*-Value ^a^
Codominant	CC	22	16.9	33	36.7	0.004
	CT	72	55.4	38	42.2	-
	TT	36	27.7	19	21.1	-
Recessive	CC	22	16.9	33	36.7	0.001
	CT/TT	108	83.1	57	63.3	-
Allele	C	116	44.6	104	57.8	0.007
	T	144	55.4	76	42.2	-

^a^*p*-value by Chi square test.

**Table 3 ijerph-17-05089-t003:** Frequencies of TXN2 (rs4821494) polymorphism among women with or without endometriosis.

TXN2 (rs4821494) Genotype		Endometriosis				
		Negative		Positive		
		N	%	N	%	*p*-Value ^a^
Codominant	GG	60	46.2	54	60.0	0.090
	GT	60	46.2	33	36.7	-
	TT	10	7.7	3	3.3	-
Recessive	GG	60	46.2	54	60.0	0.043
	GT/TT	70	53.8	36	40.0	-
Allele	G	180	69.2	141	78.3	0.035
	T	80	30.8	39	21.6	-

^a^*p*-value by Chi square test.

**Table 4 ijerph-17-05089-t004:** Frequencies of TXNRD1 (rs1128446) polymorphism among women with or without endometriosis.

TXNRD1 (rs1128446) Genotype		Endometriosis				
		Negative		Positive		
		N	%	N	%	*p*-Value ^a^
Codominant	CC	125	96.2	89	98.9	0.221
	CG	5	3.8	1	1.1	-
	GG	0	2.6	0	0	-
Recessive	CC	125	96.2	89	98.9	0.221
	CG	5	3.8	1	1.1	-
Allele	C	255	98.1	179	99.4	0.224
	G	5	1.9	1	0.6	-

^a^*p*-value by Chi square test.

**Table 5 ijerph-17-05089-t005:** The difference of allele frequency between the stage of endometriosis with the corresponding single nucleotide polymorphism in GPX4 and TXN2 genes.

Genotype	Minimal and Mild Endometriosis	Advanced Endometriosis	Patient Number	*p*-Value
GPX4 (rs713041)	-	-	-	-
CC	7 (17.5%)	26 (52%)	33	0.001 ^a^
CT	25 (62.5%)	13 (26%)	38	-
TT	8 (20.0%)	11 (22%)	19	-
TXN2 (rs4821494)	-	-	-	-
GG	21 (52.5%)	33 (66.0%)	54	0.098
GT	16 (40.0%)	17 (34.0%)	33	-
TT	3 (7.5%)	0 (0.0%)	3	-

^a^*p*-value by Chi square test.

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
