# Peer review of "Association between the Genetic Variants of Glutathione Peroxidase 4 and Severity of Endometriosis"

_ijerph, 2020, doi:10.3390/ijerph17145089_

Round 1
Reviewer 1 Report
The manuscript "Association Between the Genetic Variants of Glutathione Peroxidase 4 and Severity of Endometriosis" describes the differences in allele frequencies of GPX4, TXN1 and TXN2 between patients with endometriosis and controls. The study has a good sample size (n=220) and they found that there are differences in TXN2 and GPX4 between endometriosis patients and controls and GPX4 is also related to the severity of the disease in a Han population.
I have some minor comments:
- The sample size should be mentioned in 2.1. Study design and subjects.
- TXN1 allele frequencies were not studied between stages of the disease, probably because no differences in allele frequencies between endometriosis and controls were found. They could explain why they did not measure it in this group.
- Did they study the correlation between the severity of the disease and pain? If so, does exist a correlation with allele frequencies and pain?
- They could discuss/hypothesize a little bit more how the defect of GPX4 would affect oxidative stress and how this would affect the development of the lesions or the severity of the disease in the discussion.
- Line 222: how is GPX differently expressed in eutopic and ectopic endometrium? Is it higher/lower in eutopic? They should clarify this.
Author Response
The manuscript "Association Between the Genetic Variants of Glutathione Peroxidase 4 and Severity of Endometriosis" describes the differences in allele frequencies of GPX4, TXN1 and TXN2 between patients with endometriosis and controls. The study has a good sample size (n=220) and they found that there are differences in TXN2 and GPX4 between endometriosis patients and controls and GPX4 is also related to the severity of the disease in a Han population.
I have some minor comments:
- The sample size should be mentioned in 2.1. Study design and subjects.
Response: The sample size is mentioned in the revised manuscript.
2. TXN1 allele frequencies were not studied between stages of the disease, probably because no differences in allele frequencies between endometriosis and controls were found. They could explain why they did not measure it in this group.
Response: Thank you for the precious comment. We compare the GPX4 and TXN2 allele frequency between stages of the disease. We did not compare the TXNRD1 allele frequency between stages of endometriosis. Because the C:G allele of TXNRD1 is 179:1 and the comparison between stage of the disease is not practical for biostatistics. The explanation was added into the section of results.
3. Did they study the correlation between the severity of the disease and pain? If so, does exist a correlation with allele frequencies and pain?
Responses: Thank you for the valuable comment. We did study the correlation between the allele frequency and pain. However, the correlation is not significant by biostatistics.
4. They could discuss/hypothesize a little bit more how the defect of GPX4 would affect oxidative stress and how this would affect the development of the lesions or the severity of the disease in the discussion.
Response: We add a little more discussion about how the allele frequency or defect of GPX4 would affect oxidative stress and, consequently the development and the severity of endometriosis. The discussion part is as the following: “Since the GPX4 rs713041 SNP is associated with mortality of breast cancer patients and risk of colorectal cancer [11], the CC allele of GPX4 rs713041 SNP may increase the survival of cancer cells or ectopic endometrium by against the oxidative stress. Consequently, the CC allele is related to the occurrence and severity of endometriosis.”
5. Line 222: how is GPX differently expressed in eutopic and ectopic endometrium? Is it higher/lower in eutopic? They should clarify this.
Response: The GPX is differently expressed in eutopic and ectopic endometrium according the following reference. The results revealed that phase-dependent changes of GPx expression in the surface and glandular epithelia in the eutopic endometrium during the menstrual cycle in the fertile control. Whereas the cyclic change of GPX expression is lost in patients with endometriosis. We add further explanation into the discussion.
Ref: Ota, H.; Igarashi, S.; Kato, N.Tanaka, T. Aberrant expression of glutathione peroxidase in eutopic and ectopic endometrium in endometriosis and adenomyosis. Fertil Steril. 2000, 74, 313-8.
Reviewer 2 Report
In the manuscript entitled "Association between the genetic variants of glutathione Peroxidase 4 and severity of endometriosis", authors reported an interesting prospective case-control study to highlight the relationship between SNP of three antioxidant enzymes and endometriosis.
The work is of interest to several investigators although English quality must be improved and revised along the text.
Moreover, it would be better to explain control group features in the paragraph 2.1. Are there other significant differences among groups? Do you have data about fertility of the control group?
Finally, before it can be accepted for publication, minor revisions are required.
Author Response
In the manuscript entitled "Association between the genetic variants of glutathione Peroxidase 4 and severity of endometriosis", authors reported an interesting prospective case-control study to highlight the relationship between SNP of three antioxidant enzymes and endometriosis.
The work is of interest to several investigators although English quality must be improved and revised along the text.
Moreover, it would be better to explain control group features in the paragraph 2.1. Are there other significant differences among groups? Do you have data about fertility of the control group?
Finally, before it can be accepted for publication, minor revisions are required.
Response: Thank you for the valuable comments. The control group is simply received annual pap smear examination and donated the blood for the SNP study. We only recruited the women without records of dysmenorrhea, dyspareunia, chronic pelvic pain. The patients’ personal information is lost of connection after blood donation. Consequently, we did not have data about fertility of the control group.
Reviewer 3 Report
Minor comments,
The word "correlation" in table 5 title is confusing as this table show %. While, correlation (pearson or spearman) should be presented as correlation coefficient (r) and P-value. Please revise this part.
Author Response
The word "correlation" in table 5 title is confusing as this table show %. While, correlation (pearson or spearman) should be presented as correlation coefficient (r) and P-value. Please revise this part.
Response: Thank you for the precious comment. We change the cation of Table 5 to be the following: The correlation difference of allele frequency between the stage of endometriosis with the corresponding single nucleotide polymorphism in GPX4 and TXN2 genes.